# A multiplexed, next generation sequencing platform for high-throughput detection of SARS-CoV-2

Marie-Ming Aynaud [1,6], J. Javier Hernandez[1,2,3,6], Seda Barutcu [1,6], Ulrich Braunschweig [2], Kin Chan[1], Joel D. Pearson[1], Daniel Trcka[1], Suzanna L. Prosser [1], Jaeyoun Kim[1], Miriam Barrios-Rodiles[1], Mark Jen[1], Siyuan Song[2,4], Jess Shen[1], Christine Bruce[5], Bryn Hazlett[5], Susan Poutanen[5], Liliana Attisano [2,4], Rod Bremner [1], Benjamin J. Blencowe [2,3], Tony Mazzulli[5], Hong Han[2], Laurence Pelletier [1,3✉] & Jeffrey L. Wrana [1,3✉]

Population scale sweeps of viral pathogens, such as SARS-CoV-2, require high intensity testing for effective management. Here, we describe "Systematic Parallel Analysis of RNA coupled to Sequencing for Covid-19 screening" (C19-SPAR-Seq), a multiplexed, scalable, readily automated platform for SARS-CoV-2 detection that is capable of analyzing tens of thousands of patient samples in a single run. To address strict requirements for control of assay parameters and output demanded by clinical diagnostics, we employ a control-based Precision-Recall and Receiver Operator Characteristics (coPR) analysis to assign run-specific quality control metrics. C19-SPAR-Seq coupled to coPR on a trial cohort of several hundred patients performs with a specificity of 100% and sensitivity of 91% on samples with low viral loads, and a sensitivity of >95% on high viral loads associated with disease onset and peak transmissibility. This study establishes the feasibility of employing C19-SPAR-Seq for the large-scale monitoring of SARS-CoV-2 and other pathogens.

[1] Lunenfeld-Tanenbaum Research Institute, Mount Sinai Hospital, Toronto M5G 1X5 ON, Canada. [2] Donnelly Centre for Cellular and Biomolecular Research, University of Toronto, Toronto M5S 3E1 ON, Canada. [3] Department of Molecular Genetics, University of Toronto, Toronto M5S 1A8 ON, Canada. [4] Department of Biochemistry, University of Toronto, Toronto M5S 1A8 ON, Canada. [5] Department of Microbiology, Mount Sinai Hospital/University Health Network, Toronto M5G 1X5 ON, Canada. [6]These authors contributed equally: Marie-Ming Aynaud, J. Javier Hernandez, Seda Barutcu. ✉email: pelletier@lunenfeld.ca; wrana@lunenfeld.ca

Viral pathogens, such as SARS-CoV-2, that incorporate large numbers of asymptomatic or mild symptom patients present unique challenges for public health agencies trying to manage both travel and local spread. Physical distancing is the current major strategy to suppress spread of the disease, but with enormous socio-economic costs. However, modeling and studies in isolated jurisdictions suggest that active population surveillance through systematic molecular diagnostics, combined with contact tracing and focused quarantining can significantly suppress disease spread[1–3] and has significantly impacted disease transmission rates, the number of infected people, and prevented saturation of the healthcare system[4–7]. However, reliable systems allowing for parallel testing of tens of thousands to hundreds of thousands of patients in larger urban environments have not yet been employed. Here we describe "COVID-19 screening using Systematic Parallel Analysis of RNA coupled to Sequencing" (C19-SPAR-Seq), which is a next generation sequencing (NGS)-based platform[8] for analyzing tens of thousands of COVID-19 patient samples in a single instrument run. To enable NGS-based diagnostics we employed large numbers of control samples embedded in each run coupled to control-based Precision-Recall and predictive Receiver Operator Characteristics (coPR) analysis that assigns run-specific thresholds and quality control metrics. C19-SPAR-Seq coupled to coPR on a trial cohort of over 600 patients performed with a specificity of 100% and sensitivity of 91% on samples with low viral loads versus >95% on samples with the higher viral loads associated with disease onset and peak transmissibility. Our study thus establishes the feasibility of employing C19-SPAR-Seq for the large-scale monitoring of SARS-CoV-2 and other pathogens.

## Results

**Multiplex detection of SARS-CoV-2 using C19-SPAR-Seq.** The current gold standard diagnostic for SARS-CoV-2 is Real-Time Quantitative Polymerase Chain Reaction (RT-qPCR), which is not readily adaptable to large-scale population testing[9]. To establish a population-scale testing platform we designed a SPAR-Seq multiplex primer mix v1 that targets RNA-dependent RNA polymerase (*RdRP*), Envelope (*E*), Nucleocapsid (*N*), and two regions of the Spike (*S*) gene that correspond to the receptor-binding domain (RBD) and the polybasic cleavage site (PBS) (Fig. 1a, Supplementary Table 1 and Supplementary Data 1). The latter two are SARS-CoV-2-specific regions that capture five key residues necessary for ACE2 receptor binding (*Srbd*) and the furin cleavage site (*Spbs*) that is critical for viral infectivity[10,11]. Thus, the RdRP-specific primers could produce an amplicon from SARS-CoV-1 that can be readily distinguished based on sequence analysis, while the Spike-specific primers, targeting the RBD and Polybasic site regions, would distinguish a SARS-CoV-2 infection. For quality control, we targeted Peptidylprolyl Isomerase B (*PPIB*). Current standard testing strategies for viral pathogens employ gene-specific primers in "all-in-one" qRT-PCR reactions that could in principle be adapted to incorporate barcodes into gene-specific primers. However, to allow for rapid adaptation to test for novel and multiple pathogens, and/or profiling host responses we used a generic oligo-dT and random hexamer primed reverse transcription step followed by multiplex PCR and barcoding in a rapid, readily automated format we call "COVID-19 screening using Systematic Parallel Analysis of RNA coupled to Sequencing" or C19-SPAR-Seq (Fig. 1b, Supplementary Table 1 and Supplementary Data 1). Although cost is often cited as a concern for NGS-based testing, our platform is cost effective with retail material costs ranging from USD ~$9 to $6 for 500 versus 10,000 sample batch sizes, respectively (Supplementary Data 2).

To assess C19-SPAR-Seq performance, we assembled a proof-of-concept (PoC) cohort of 19 archival Nasopharyngeal (NASOP)

swab eluents from the Toronto University Health Network-Mount Sinai Hospital clinical diagnostics lab (Supplementary Data 3), 17 of which were positive for SARS-CoV-2. Viral load in these archival samples was quantified using the clinically approved TaqMan-based SARS-CoV-2 RT-qPCR detection kit ('BGI', see the "Methods" section), which identified five SARS-CoV-2$^{low}$ (Ct > 25), seven SARS-CoV-2$^{medium}$ (Ct between 20 and 25), and five SARS-CoV-2$^{high}$ (Ct < 20) patients (Supplementary Data 3). After confirming the efficiency of multiplex v1 primer pairs using a SARS-CoV-2$^{high}$ sample (LTRI-18, Ct < 20; Supplementary Fig. 1), we performed C19-SPAR-Seq using HEK293T RNA as a negative control ($n = 2$), and serial dilutions of synthetic SARS-CoV-2 RNA (Twist) as positive controls ($n = 5$). Pooled sequence data was demultiplexed to individual samples prior to mapping to amplicon sequences. C19-SPAR-Seq was sensitive in detecting as low as 12.5 copies/µL of E, Srbd, and Spbs amplicons from Twist RNA (Fig. 1c, left panel). In patient samples, *PPIB* was present in all samples, and all viral targets were robustly detected in high/medium load samples, with reduced detection of *E* and *RdRP* genes in low samples (Fig. 1c, right panel).

**Development of a C19-SPAR-Seq diagnostic platform to detect SARS-CoV-2.** To establish a diagnostic platform, we performed C19-SPAR-Seq on a larger test development cohort of 24 COVID-19 positive and 88 negative archival patient samples ($n = 112$; Supplementary Data 4). The SARS-CoV-2 RNA standard curve showed a linear relationship between total viral reads and estimated viral copy numbers (Supplementary Fig. 2a). Negative patient samples had low viral reads (median of 4; range 0–55) compared to positive samples (median of 5899; range 2–253,956 corresponding to 18–705,960 amplicon reads per million reads per sample) (Fig. 2a). C19-SPAR-Seq read counts tracked inversely with qRT-PCR Ct values for *RdRP*, *E*, and *N* genes quantified in the diagnostic lab using the Seegene Allplex$^{TM}$ assay (see the "Methods" section) (Fig. 2b). Unsupervised clustering showed that the controls performed similarly to the PoC cohort (Fig. 2c), as did the positive and negative patient samples, with two exceptions: clinical samples LTRI042 and LTRI050, which displayed background signal, and corresponded to samples with extreme Ct values in only one viral gene (*N* gene, Ct > 38; Supplementary Data 4). ROC analysis using total viral reads (Fig. 2d) showed excellent performance with an area under the ROC curve (AUC) of 0.969. Using PROC, the point on the ROC curve that minimizes the distance to (0,1)[12], defined a total viral read cut-off of 116 for calling a positive sample and yielded a sensitivity of 92% (95% confidence interval; CI of 73–99%), specificity of 100% (CI: 95–100%), and overall accuracy of 98% (Fig. 2d). Using Youden parameters that maximize sensitivity and specificity defined a viral cutoff of 26 and yielded better sensitivity (96%), but lower specificity (95%) and accuracy (96%). Other than the two positive samples mentioned above that possessed extremely low levels of viral RNA (Ct 38 and 40), all other positive samples were above the C19-SPAR-Seq viral threshold limit, indicating that the lower limit of sensitivity in the CI is dictated by these samples that lie at the border of the detection limit of the diagnostic lab test. Thus, C19-SPAR-Seq robustly detects SARS-CoV-2 transcripts, correlates with Ct values from clinical diagnostic tests, and displays excellent performance in distinguishing positive and negative samples.

**An internal control-based classifier to assess patient samples.** Robust application of C19-SPAR-Seq as a diagnostic tool requires assigning thresholds for both viral RNA detection, as well as host RNA for filtering poor quality samples. In qRT-PCR diagnostics,

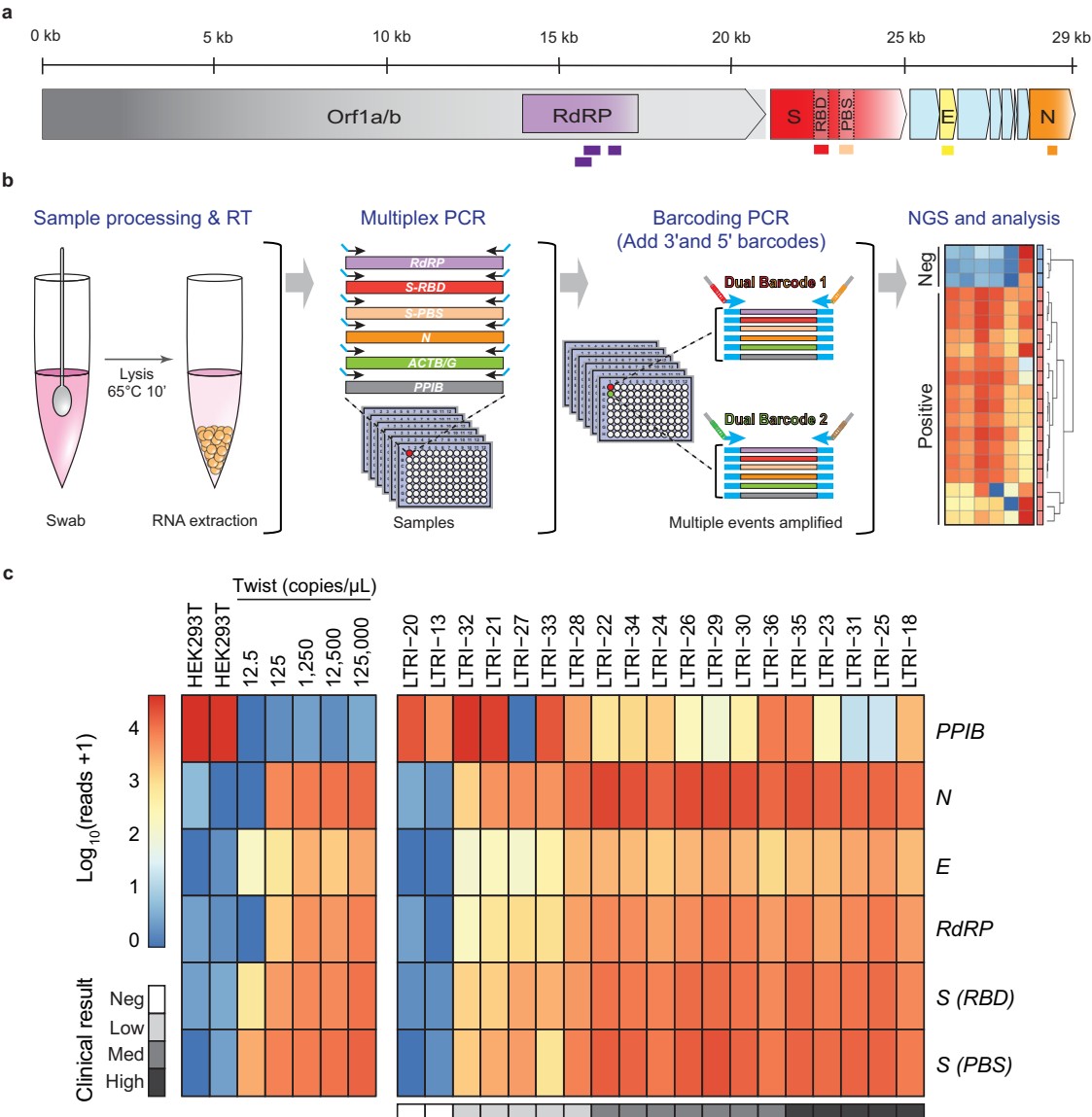

**Fig. 1 Application of C19-SPAR-Seq to detect SARS-CoV-2. a** Schematic representation of the SARS-CoV-2 with the five regions targeted for multiplex C19-SPAR-Seq indicated: *RdRP* (purple), *S receptor-binding domain* (*Srbd*) (red), *S polybasic cleavage site* (*Spbs*) (light red), *E* (yellow), and *N* (orange). **b** Schematic of the C19-SPAR-Seq strategy for detecting SARS-CoV-2. cDNA is synthesized using reverse transcriptase (RT) from RNA extracted from clinical samples, subjected to multiplex PCR, then barcoded, pooled, and analyzed by next generation sequencing (NGS). **c** Analysis of archival NASOP swab eluents by C19-SPAR-Seq. A Proof-of-Concept (PoC) cohort (n = 19) was analyzed by C19-SPAR-Seq and read numbers for each of the indicated amplicons are presented in a heatmap. Control samples (HEK293T, synthetic SARS-CoV-2 RNA) are represented in the left panel, while the right panel shows unsupervised 2D hierarchical clustering of results from negative (blue) and positive (red) patients.

external validation studies and rigorous standard operating procedures establish pre-defined cutoffs for sample quality and positive versus negative assignment (Seegene, see the "Methods" section); BGI (see the "Methods" section) However, in scalable, massively parallel, multiplexed NGS assays, variability in sample numbers and flow cell loading can create run-to-run variations in read numbers, while index-mismatching[13], as well as trace cross-contamination events can create technical noise that are challenging to control. Furthermore, external validation strategies create a laborious path to adapt and test new multiplex designs to SARS-CoV-2, additional respiratory pathogens, or host responses. We therefore exploited the throughput of C19-SPAR-Seq to include in every run a training set of large numbers of controls that can be exploited to define cutoffs tailored to each C19-SPAR-Seq run (Fig. 3a). To define quality metrics, we computed precision-recall (PR) curves for classifying control samples as either negative (H₂O

blanks), or positive for any anticipated amplicon (HEK293T for PPIB or synthetic SARS-CoV-2 RNA for viral amplicons) and calculated the highest F1 score, which is the harmonic mean of PR and a common measure of classifier accuracy (Fig. 3b). When mapped onto a ROC curve this corresponded to the region closest to perfect sensitivity and specificity (0, 1) (Supplementary Fig. 2b). To define the threshold for identifying SARS-CoV-2-positive cases, we next analyzed the embedded standard curve of synthetic SARS-CoV-2 RNA. This displayed a linear relationship over four orders of magnitude that extended to lower limits of detection indistinguishable from background reads from HEK293T cells (Fig. 2a and Supplementary Fig. 2a), thus allowing us to identify the viral read count in each C19-SPAR-Seq run that most accurately distinguishes positive from negative (Fig. 3a). To identify this threshold, we computed PROC01, which optimizes negative predictive value (NPV) and positive predictive value (PPV)[12] and

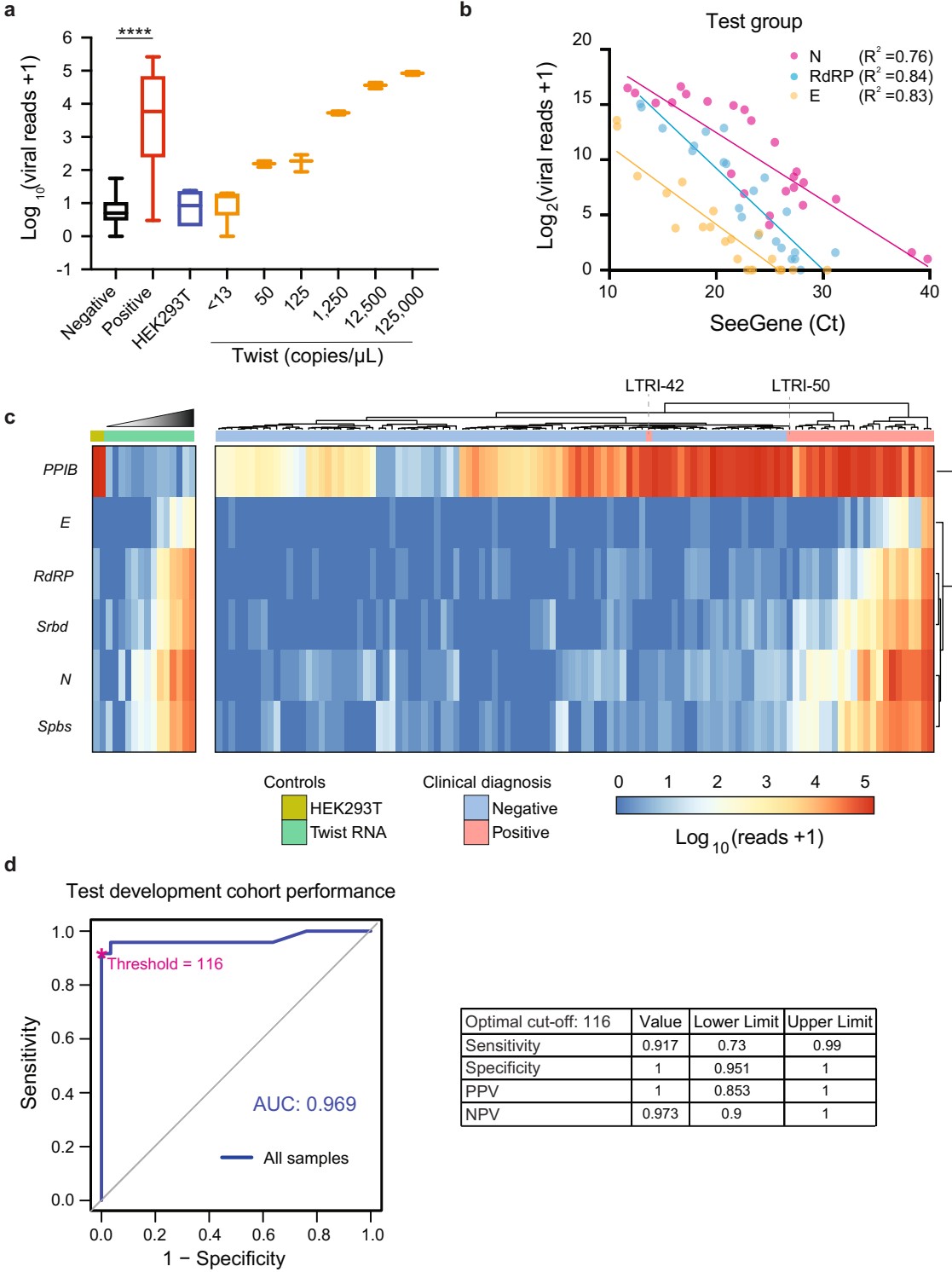

**Fig. 2 Performance of C19-SPAR-Seq in detecting SARS-CoV-2. a** C19-SPAR-Seq of the test development cohort was performed and total viral reads+1 ($\log_{10}$) (*Y*-axis) are plotted for negative ($n = 88$, black) and positive ($n = 24$, red) patient samples, HEK293T RNA ($n = 6$, blue), and the indicated serial dilutions of synthetic SARS-CoV-2 RNA ($n = 2$-6, orange). For each group, the median, lower and upper confidence limits for 95% of the median are plotted. Whiskers are minimum and maximum values. Two-tailed unpaired *t*-test of negative versus positive samples (****$p = 1.67 \times 10^{-8}$). **b** C19-SPAR-Seq reads for the indicated gene in each patient sample were compared to Ct values obtained by the clinical diagnostics lab using the 'Seegene' Allplex assay. **c** Heatmap of C19-SPAR-Seq results. Read counts for the indicated target amplicons in control samples ($n = 16$; left) and patient samples ($n = 112$; right) are plotted according to the scale, and sample types labeled as indicated. Samples are arranged by hierarchical clustering with euclidean distance indicated by the dendrogram on the top, which readily distinguishes positive from negative samples. **d** Performance of C19-SPAR-Seq. ROC analysis on patient samples was performed using clinical diagnostic results (Seegene Allplex qRT-PCR assay, Supplementary Data 4) and total viral reads for patient samples ($n = 112$). AUC (area under the curve) scores are indicated on the graph (left), with statistics at the optimal cutoff as indicated (right).

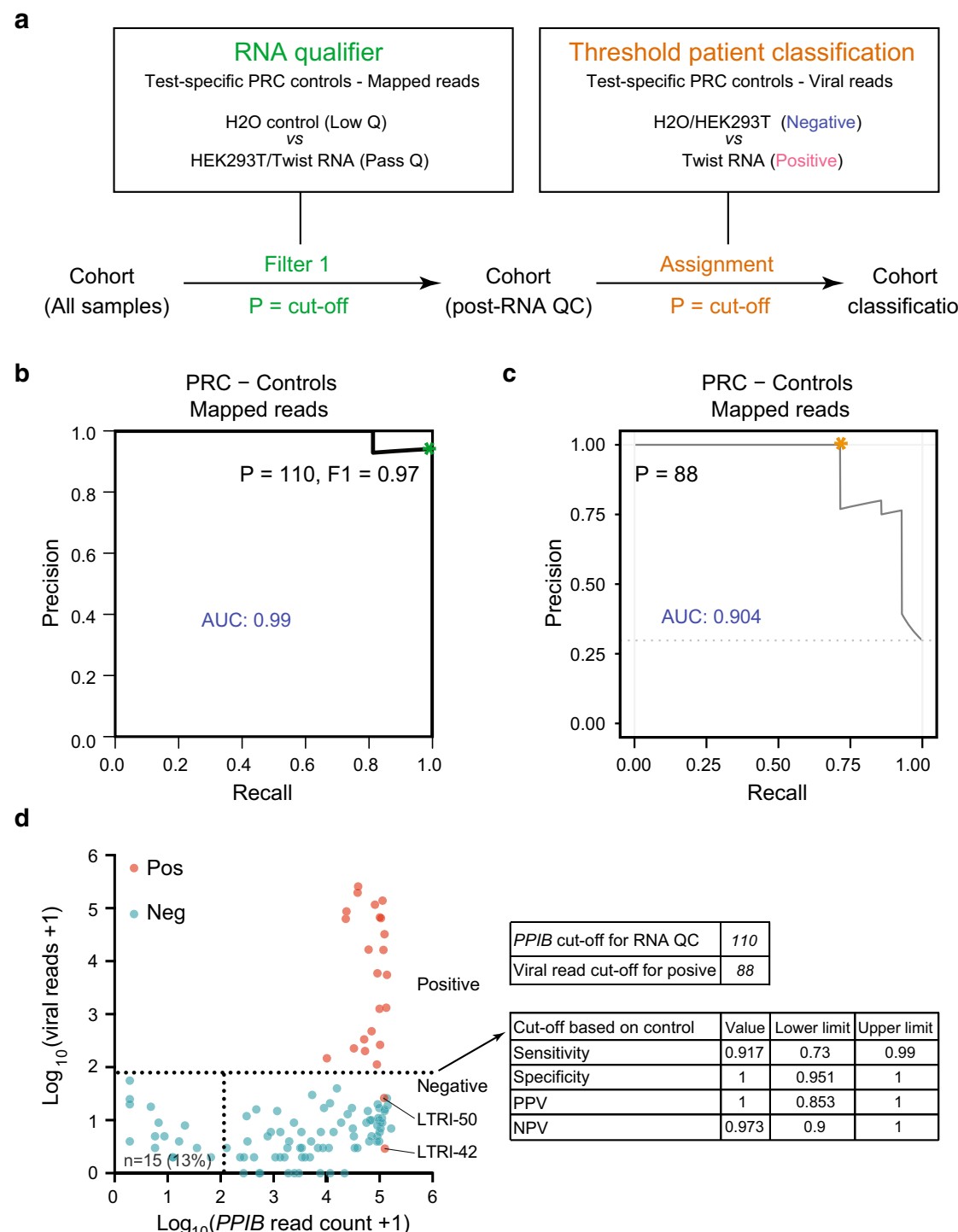

**Fig. 3 Performance of C19-SPAR-Seq in detecting SARS-CoV-2 using control-based classifier. a** Schematic of the control-based cut-off procedure for RNA quality and viral threshold by coPR analysis. **b** Thresholding sample quality. coPR analysis on control samples: PRC of control samples for accurate detection of mapped reads are plotted. The optimal precision and recall read cut-off associated ($P = 110$) with the highest F1 (0.97) score, and AUC (area under the curve) are indicated in the PR plot. **c** Threshold for classification of positives in the test cohort. Optimum cut-off for viral threshold is calculated by PROC01 using clinical diagnosis and total viral reads and plotted on the precision-recall curve. **d** Threshold assignments for sample quality and classification. Total viral reads $+1$ ($Y$-axis) are plotted against $PPIB$ reads $+1$ ($X$-axis) for positive (red) and negative (blue) patient samples. coPR-based RNA-QC filter and viral read filter are shown as indicated. Assay statistics using coPR thresholding are listed (right).

defined a point (88 viral reads) close to perfect PR (Fig. 3c) and sensitivity and specificity on the ROC curve (Supplementary Fig. 2c). Importantly, these methods control for run-specific variables by employing training sets that are embedded in every C19-SPAR-Seq run.

We next mapped the control-based cutoffs onto our patient SPAR-Seq data (Fig. 3d). This showed 15 of these archival samples had low $PPIB$ counts that may be due to lost RNA integrity upon repeated freeze–thaw cycles (Fig. 3d and Supplementary Data 4), a variability we also observed in the

PoC cohort (Fig. 1c). Of note, C19-SPAR-Seq performance was not affected by filtering poor quality samples (AUC = 0.970; Supplementary Fig. 2d). Furthermore, using PROC01 thresholding of viral reads identified 22/24 positives with no false positives (Fig. 3d). This yielded an overall test performance of 92% sensitivity, 100% specificity, and 98% accuracy (Fig. 3d and Supplementary Tables 2, 3). This is similar to the observed performance of C19-SPAR-Seq on clinical samples quantified by ROC analysis (Fig. 2d and Supplementary Fig. 2d, respectively). Thus, an extensive array of internal reference samples is effective as an embedded training set for implementing a control-based PR/PROC classifier (coPR) that is tailored to each C19-SPAR-Seq run.

**Negative samples create noise in C19-SPAR-Seq.** To validate our C19-SPAR-Seq platform we established a pilot cohort of 378 samples that contains 89 positive samples collected in May of 2020. We first screened for positivity using the clinically approved BGI SARS-CoV-2 kit (see the "Methods" section) which showed 52 samples were positive with >4 viral copies/µL (Supplementary Data 5 and Supplementary Table 4). Of the 37 failed samples, 86% had very low viral RNA (only 1 or 2 of the 3 genes detected and/or Ct > 35 on the 'Seegene' platform) that may have lost integrity upon storage. Indeed, comparison of Ct values for RdRP detection showed an overall increase of four cycles in these archived samples (Supplementary Fig. 3a), despite the high sensitivity of the BGI platform[14]. The cohort also contained 289 negative samples collected prior to Ontario's[15] first confirmed COVID-19-positive case in January 20, 2020, and 1 negative sample collected in May 2020 (Supplementary Data 5), and included broncho-alveolar lavages (BALs) and NASOP swabs. Surprisingly, the detection of human RNA dropped substantially to a median of 29 (range 0–41,874), compared to 15,058 (range 2–170,870) in the original test cohort. coPR filtering (Supplementary Fig. 3b), marked 50% of samples as inconclusive compared to 13% in the test cohort (Supplementary Fig. 3c), despite similar distribution of raw reads per sample (Supplementary Fig. 3d), while mapping rates in the PoC, test and pilot cohorts, progressively declined to as low as 0.1% (Supplementary Fig. 3e). To understand this collapse we analyzed unmapped reads and found that >90% were consumed by non-specific amplification products (NSAs; Supplementary Fig. 4a) that comprised complex chimeric combinations of many viral and human primers (Supplementary Fig. 4a, b). For example, RdRP and PPIB contributed to 4 of the top 5 NSAs (NSA1–4), and 2 had a spurious sequence (NSA4, 5). Indeed, analysis of C19-SPAR-Seq PoC, test and pilot libraries using a Bioanalyzer, showed that as cohort size and number of negatives increased, NSAs were more apparent, and dominated the pilot library (Supplementary Fig. 4c). This suggests that NSAs, enriched in negative samples (3.7-fold increase in the pilot cohort), clog the NGS pipeline as sample numbers rise (Supplementary Table 4). This has serious implications for deploying an NGS platform in a population-scale COVID-19 surveillance strategy and highlights the importance of using large-scale cohorts during the development of multiplex testing platforms.

**Analyzing an extended cohort using an optimized multiplex panel v2.0.** SARS-CoV-2 RNA concentration spans a large dynamic range, such that spike-in mutant amplicons which have been suggested to improve performance of NGS-based strategies[16] might interfere with detection of COVID-19-positive cases with low viral reads. Therefore, we instead used our NSA data to create multiplex panel v2.0 (see the "Methods" section) that removed primers yielding NSAs by targeting a distinct region of RdRP,

removing E and N genes, and switching to primers that amplify intron spanning regions of the ACTB and ACTG genes (Supplementary Table 1, Supplementary Data 1 and Supplementary Fig. 1). We extended the pilot cohort to 663 samples that included 98 confirmed positives and performed C19-SPAR-Seq, which showed targeted amplicons were the predominant product generated by multiplex panel v2.0 (Supplementary Fig. 5a), and mapping percentages were restored to test cohort levels (Supplementary Fig. 5b). Total viral read distributions for multiplex panel v2.0 showed good separation in clinically positive samples (Fig. 4a and Supplementary Fig. 5c), while applying coPR thresholding (Supplementary Fig. 5d) identified 121 samples as inconclusive (Fig. 4a), all of which were older, pre-COVID19 material. Of these, 112 were BALs (40% of all BALs), 1 was a bronchial wash (BMSH), and only 8 were NASOPs (1.8% of all NASOPs) (Supplementary Data 6). Furthermore, analysis of 10 BAL samples below the QC threshold revealed little or no RNA, contrasting BALs with moderate levels of ACTB/G transcripts (representative examples in Supplementary Fig. 6a), and BAL ACTB/G read distributions were much lower than NASOPs (Supplementary Fig. 6b). This suggests that archival BALs suffered from substantive sample degradation and also highlights how coPR-based thresholding successfully identifies poor quality samples and readily adapts to the use of distinct primer sets.

Next, we analyzed viral reads, which had a broad range in positive samples (median = 680.5 reads per sample, range 0–200,850; Fig. 4a and Supplementary Fig. 5c). Two-dimensional clustering showed background SARS-CoV-2 products in negative samples were low to undetectable, and ACTB typically yielded higher reads than ACTG, likely reflecting their differential expression (Fig. 4b). Positive samples were generally well separated, although some distinct clusters with lower SARS-CoV-2 reads were apparent (Fig. 4b and Supplementary Fig. 5e). Indeed, total read distributions in positive samples displayed biphasic distribution (Supplementary Fig. 5e), similar to observations made from RT-qPCR analyses of ~4000 positive patients[17]. Since the early rapid increase in SARS-CoV-2 viral load at symptom onset is followed by a long tail of low viral load during recovery[18,19], this biphasic distribution could reflect patients in distinct phases of the disease. We also assessed viral amplicon sequences which matched the SARS-CoV-2 reference (MN908947.3[20]) and found no variants (Supplementary Fig. 5f). Since neutralizing antibodies are generally thought to target the critical region of the RBD analyzed here[15], these results suggest the emergence of variant strains that might bypass acquired immunity is not a major feature of SARS-CoV-2. In addition, this supports the notion that biologic therapies targeting the RBD may show broad activity in the population.

We next compared performance of multiplex panel v2.0 to v1.0 using the embedded controls, which showed similar performance (AUC = 0.90, Supplementary Fig. 5g versus 0.92, Supplementary Fig. 2c, respectively), with coPR yielding an optimal read cutoff of >16 total viral reads (Supplementary Fig. 5f) that corresponded to a technical sensitivity of 3 viral copies/µL (Supplementary Fig. 6c). coPR thus identified 82 positive samples (Fig. 4a and Supplementary Data 6), all of which were BGI-confirmed cases, to give an overall sensitivity of 84%, specificity of 100%, and accuracy of 97% (Supplementary Table 5 and Fig. 4a). Importantly, total viral reads tracked with BGI Ct values (Fig. 4c), and for samples with Ct < 35 (corresponding to ~12 viral copies/µL of specimen), sensitivity was similar to the test cohort at 91%. However, for samples with Ct between 35 and 37 (4–12 viral copies/µL) sensitivity dropped markedly to 44% (Supplementary Table 5 and Fig. 4a), while at higher viral loads (Ct = 25 or ~8400 viral copies/µL) sensitivity rose to 100% (Fig. 4c). ROC analysis of actual C19-SAR-Seq performance yielded an AUC of 0.96, sensitivity of 87% and specificity of 100%, similar to coPR (Fig. 4d), while individual

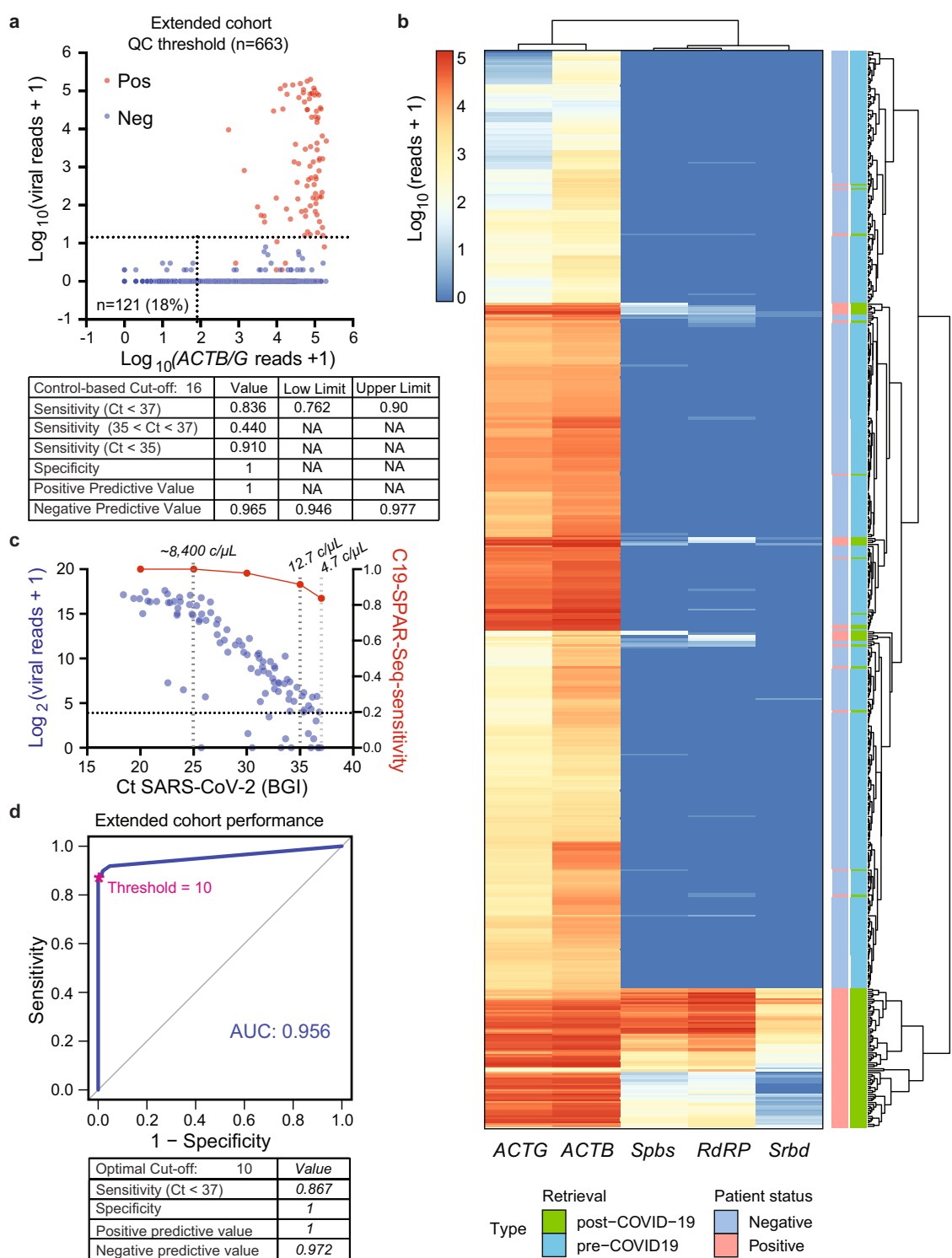

**Fig. 4 C19-SPAR-Seq of a large patient cohort. a** C19-SPAR-Seq on an extended patient cohort. coPR thresholds for sample quality and classification of a 663 patient cohort of negative (blue) and positive (red) specimens are shown as in Fig. 2a. Performance metrics with 95% confidence intervals for sample classification according to coPR thresholding are shown in the table. NA not applicable. **b** Heatmap of C19-SPAR-Seq results. Read counts for the indicated target amplicons in the filtered set of samples ($n = 542$) are plotted according to the scale, and sample types labeled as indicated. Samples are arranged by hierarchical clustering with euclidean distance indicated by the dendrogram on the right. **c** Scatter plot of total viral reads+1 (left Y-axis, blue) versus Ct values of positive samples ($n = 98$, BGI) (X-axis). C19-SPAR-seq sensitivity at the indicated Ct values is overlaid (right Y-axis, red). Gray dashed lines indicate average copies/μL (c/μL). **d** ROC curve analysis. ROC curves were processed on filtered samples ($n = 542$). AUC scores are indicated for filtered samples (blue; left) with corresponding performance statistics for the optimal cut-off indicated below.

amplicons each underperformed total viral reads (AUC: 0.85–0.94; Supplementary Fig. 6d). Our cohort was biased for samples with very low to low viral loads, which represents a small portion of the COVID-19 population[17]. This bias could lead to an underestimate of the sensitivity of C19-SPAR-Seq in the context of a large-scale population, so we mapped our sensitivity data at distinct viral loads onto the population distribution of viral loads obtained from ~4000 positive patients[17]. This showed a projected C19-SPAR-Seq sensitivity of ~97% for patients displaying >10,000 viral copies/mL (Supplementary Fig. 6e), which encompasses ~90% of the positive population. Altogether, these results demonstrate that in high patient sample loads comprised of predominantly negative samples, C19-SPAR-Seq using coPR displays 100% specificity and >95% sensitivity at viral loads typically observed in populations.

## Discussion

Systematic population-scale testing has been identified as an important tool in managing pandemics such as SARS-CoV-2, where large numbers of infected individuals display mild or no symptoms yet transmit disease. The scalable throughput of C19-SPAR-Seq combined with its excellent sensitivity and specificity at reasonable cost make it well-suited for this role. Data generated by large-scale routine testing of local and larger communities, with different interaction levels would provide valuable epidemiologic information on mechanisms of viral transmission, particularly when coupled to multiplex panels targeting regions of sequence variance currently in development. Indeed, while we detected no variants in our positive samples collected in the Spring of 2020, the S-RBD and S-PBS amplicons will detect the newly emergent N501Y and P681H variants[21,22]. In addition, the C19-SPAR-Seq platform can be readily adapted to incorporate panels tracking multiple pathogens, as well as host responses. C19-SPAR-Seq quantitation would also facilitate real-time tracking of viral load dynamics in populations that may be associated with COVID-19 expansion or resolution phases[18]. Although C19-SPAR-Seq is dependent on centralized regional facilities, it is readily coupled to saliva-based, at-home collection that exploits extensive transport infrastructure and industrialized sample processing to enable frequent widespread testing.

## Methods

**Samples collection**. Patient samples (Supplementary Data 3–6) were obtained from the Department of Microbiology at Mount Sinai Hospital. Patient samples used in this study were approved by the Mount Sinai Hospital (MSH) Research Ethics Board (REB): MSH REB Study #20-0078-E 'Use of known COVID-19 status tissue samples for development and validation of novel detection methodologies'. The patient samples were de-identified prior to transfer from the Mount Sinai Hospital Microbiology Department to our research staff. The samples were excess to clinical need and considered residual samples which do not require informed consent for the secondary use of the de-identified biological materials utilized in this study. Patient samples were obtained as part of routine diagnostic testing.

**Total RNA extraction**. A step-by-step protocol describing the patient RNA extraction protocol can be found at Protocol Exchange[23]. Total RNA was extracted by using the Total RNA extraction kit (Norgen Biotek kit, Cat. #7200) for the samples in Supplementary Data 3 following the manufacturers guidelines. For all other samples (Supplementary Data 4–6), total RNA was purified in 96-well plates using RNAclean XP beads (Beckman, A66514) and a customized protocol. Briefly, 75.25 μL of patient swabs in transfer buffer were mixed with 14.5 μL of 10× SDS lysis buffer (1% SDS, 10 mM EDTA), 48 μL of 6 M GuHCl, and 7.25 μL proteinase K (20 mg/mL, ThermoFisher, 4333793), incubated at room temperature for 10 min and heated at 65 °C for 10 min prior to the addition of 145 μL of beads. Beads were washed twice in 70% ethanol using a magnetic stand and then RNA eluted into a 30 μL Resuspension buffer supplied with the kit. RNA quality was assessed using a Bioanalyzer (5200 Agilent Fragment Analyzer). HEK293T RNA was extracted using the Total RNA extraction kit (Qiagen). Synthetic Twist SARS-CoV-2 RNA (Twist Bioscience #102024-MN908947.3) was used as positive control.

**Reverse transcription (RT)**. A step-by-step protocol describing the reverse transcription protocol can be found at Protocol Exchange[23]. Total RNA was reverse transcribed using SuperScript™ III Reverse Transcriptase (Invitrogen) in 5× First-Strand Buffer containing DTT, a custom mix of Oligo-dT (Sigma) and Hexamer random primers (Sigma), dNTPs (Genedirex), and Ribolock RNase inhibitor (ThermoScientific). We followed the manufacturer's protocol. Each reaction included: 0.5 μL Oligo-dT, 0.5 μL hexamers, 4 μL purified Total RNA, 1 μL dNTP (2.5 mM each dATP, dGTP, dCTP and dTTP), *quantum satis* (*qs*) 13 μL RNase/DNase free water. Samples were incubated at 65 °C for 5 min, and then placed on ice for at least for 1 min. The following was added to each reaction: 4 μL 5× First-Strand Buffer, 1 μL 0.1 M DTT, 1 μL Ribolock RNase Inhibitor, 1 μL of SuperScript™ III RT (200 units/μL) and then mixed by gently pipetting. Samples were incubated at 25 °C for 5 min, 50 °C for 60 min, 70 °C for 15 min and then stored at 4 °C.

**TaqMan-based RT-qPCR detection**. A Real-Time Fluorescent RT-PCR kit from 'BGI' was used according to manufacturer's instructions (Cat no. MFG030010, BGI Genomics Co. Ltd. Shenzhen, China). Experiments were carried out in a 10 μL reaction volume in 384-well plates, using 3 μL of sample (LTRI patient samples or Twist RNA), and were analyzed using a Bio-Rad CFX384 detection system (Supplementary Data 3, 5, 6). Real-time Fluorescent RT-PCR results from 'Seegene' assay were provided by the Department of Microbiology diagnostic lab at Mount Sinai Hospital (Supplementary Data 4–6) (AllplexTM 2019-nCoV Assay, version 2.0, Cat no. RP10250X/RP10252W, Seegene).

**C19-SPAR-Seq primer design and optimization**. Optimized multiplex PCR primers for SARS-CoV-2 (*N*, *S*, *E* and *RdRP*) and human genes (*PPIB* and *ACTB/G*) were designed using the SPAR-Seq pipeline[8], with amplicon size >100 bases (see Supplementary Table 1 and Supplementary Data 1). For the *S* gene, two regions were monitored, the *S receptor-binding domain* (*Srbd*), and *S polybasic cleavage site* (*Spbs*). The Universal adapter sequences used for sequencing were F: 5′-acactctttccctacacgacgctcttccgatct and R: 5′-gtgactggagttcagacgtgtgctcttccgatct). Primers were optimized to avoid primer–dimer and non-specific multiplex amplification. To assess the primers sensitivity and specificity, we performed qPCR (SYBR green master mix, BioApplied) on cDNA prepared from patient samples. Each primer was used at 0.1 μM in qPCR reaction run on 384-well plates using Biorad CFX 384 detection system. The thermal cycling conditions were as follows: one cycle at 95 °C for 2 min, and then 40 cycles of 95 °C for 15 arcsec, 60 °C for 15 arcsec, 72 °C for 20 arcsec, followed by a final melting curve step.

**Multiplexing PCR**. A step-by-step protocol describing the multiplex PCR protocol can be found at Protocol Exchange[23]. The multiplex PCR reaction was carried out using Phusion polymerase (ThermoFisher). The manufacturer's recommended protocol was followed with the following primer concentrations: all primers (*N*, *Spbs*, *Srbd*, *E*, *RdRP*, and *PPIB*) were at 0.1 μM for the PoC cohort (Supplementary Data 3), SARS-CoV-2 primers (*N*, *Spbs*, *Srbd*, *E* and *RdRP*) were at 0.05 μM, and *PPIB* primer was at 0.1 μM for the test and pilot cohort (Supplementary Data 4 and 5), all primers (*Spbs*, *Srbd*, *RdRP* and *ACTB/G*) were at 0.05 μM for the extended cohort (Supplementary Data 6). For each reaction: 5 μL 5× Phusion buffer, 0.5 μL dNTP (2.5 mM each dATP, dGTP, dCTP, and dTTP), 0.25 μL for each human primers (10 μM), 0.125 μL for each SARS-CoV2 primers (10 μM), 2 μL of cDNA, 0.25 μL Phusion Hot start polymerase, *qs* 25 μL RNase/DNase free water. The thermal cycling conditions were as follows: one cycle at 98 °C for 2 min, and 30 cycles of 98 °C for 15 arcsec, 60 °C for 15 arcsec, 72 °C for 20 arcsec, and a final extension step at 72 °C for 5 min and then stored at 4 °C for the PoC and extended cohorts (Supplementary Data 3, 6), one cycle at 98 °C for 2 min, and 35 cycles of 98 °C for 15 arcsec, 60 °C for 15 arcsec, 72 °C for 20 arcsec, and a final extension step at 72 °C for 5 min and then stored at 4 °C for the test and pilot cohorts (Supplementary Data 4 and 5).

**Barcoding PCR**. A step-by-step protocol describing the barcoding PCR protocol can be found at Protocol Exchange[23]. For multiplex barcode sequencing, dual-index barcodes were used[8]. The second PCR reaction on multiplex PCR was performed using Phusion polymerase (ThermoFisher). For each reaction: 4 μL 5× Phusion buffer, 0.4 μL dNTP (2.5 mM each dATP, dGTP, dCTP, and dTTP), 2 μL Barcoding primers F + R (pre-mix), 4 μL of multiplex PCR reaction, 0.2 μL Phusion polymerase, *qs* 20 μL RNase/DNase-free water. The thermal cycling conditions were as follows: one cycle at 98 °C for 30 arcsec, and 15 cycles of 98 °C for 10 arcsec, 65 °C for 30 arcsec, 72 °C for 30 arcsec, and a final extension step at 72 °C for 5 min and stored at 4 °C.

**Library preparation and sequencing**. A step-by-step protocol describing the library preparation and sequencing protocol can be found at Protocol Exchange[23]. For all libraries, each sample was pooled (7 μL/sample) and library PCR products were purified with SPRIselect beads (A66514, Beckman Coulter). The PoC, test, and pilot cohorts were purified as follows: ratio 0.8:1 (beads:library), and the extended cohort with 1:1 (beads:library) (Beckman Coulter). Due to NSA products in the fragment analyzer profile (Supplementary Fig. 3c) in the test cohort and pilot cohort, we performed size selection purification (220–350 bp) using the Pippin Prep system (Pippin HT, Sage Science). Library quality was assessed with the 5200 Agilent Fragment Analyzer (ThermoFisher) and Qubit 2.0 Fluorometer

(ThermoFisher). All libraries were sequenced with MiSeq or NextSeq 500 (Illumina) using 75 bp paired-end sequencing.

**COVID-19 (C19-)SPAR-Seq platform**. A step-by-step protocol describing the COVID-19 (C19-)SPAR-Seq platform protocol can be found at Protocol Exchange[23]. Our Systematic Parallel Analysis of Endogenous RNA Regulation Coupled to Barcode Sequencing (SPAR-Seq) system[8] was modified to simultaneously monitor COVID-19 viral targets and additional controls by multiplex PCR assays. For barcode sequencing, unique, dual-index C19-SPAR-Seq barcodes were used. Unique reverse 8-nucleotide barcodes were used for each sample, while forward 8-based barcodes were used to mark each half (48) of the samples in 96-well plate to provide additional redundancy. These two sets of barcodes were incorporated into forward and reverse primers, respectively, after the universal adaptor sequences and were added to the amplicons in the second PCR reaction. The C19-SPAR-Seq analysis pipeline with the algorithms used is explained in detail in Supplementary Fig. 7 with additional analytical tools described in Supplementary Fig. 8 and below in the "Methods" sections. Computational requirements for the demultiplexing step is 32 GB RAM and minimum 1 GB network infrastructure, with a Linux-operating system.

**Demultiplexing and mapping**. Illumina MiSeq sequencing data was demultiplexed based on perfect matches to unique combinations of the forward and reverse 8 nucleotide barcodes. Full-length forward and reverse reads were separately aligned to dedicated libraries of expected amplicon sequences using bowtie[24] with parameters –best -v 3 -k 1 -m 1. Read counts per amplicon were represented as reads per million or absolute read counts. The scripts for these steps are available at https://github.com/UBrau/SPARpipe[25].

**Filtering of low-input samples**. To remove samples with low amplified product, likely reflecting low input due to inefficient sample collection or degradation, before attempting to classify, we computed precision-recall curves for classifying control samples into 'low amplification' and 'high amplification' based on reads mapped to RNA amplicons but ignoring mapping to genomic sequence, if applicable. The former group comprised all controls in which individual steps were omitted (H2O controls) and the latter comprised HEK293T as well as synthetic SARS-CoV-2 RNA controls. For each PoC, test, pilot, extended runs, we obtained the total mapped read threshold (including reads mapping to both human and viral amplicons) associated with the highest F1 score, representing the point with optimal balance of precision and recall. Samples with reads lower than this threshold were removed from subsequent steps. Scripts for this step are available at https://github.com/UBrau/ModelPerformance[26].

**SARS-CoV2-positive sample classification**. To assign positive and negative samples, we used negative (H2O and HEK293T) and positive (synthetic SARS-CoV-2 RNA dilutions) internal controls for each run and calculated optimum cut-offs for viral reads (total reads mapping to all three viral amplicons) by PROC which defines the threshold for optimum positive predictive value (PPV) and negative predictive value (NPV) for diagnostic tests. Thus, a sample was labeled positive if it had viral reads above the viral read threshold; negative if it had viral reads below the viral read threshold and human reads above the mapped read threshold; and inconclusive if it had both viral and human reads below the respective thresholds.

**Sample classification by heatmap clustering**. Heatmap and hierarchical clustering of viral and control amplicons, $\log_{10}(\text{mapped reads} + 1)$, was used to analyze and classify all samples. Samples with a total mapped read count lower than the RNA QC threshold were labeled as inconclusive and removed before the analysis. Known positive (high, medium, and low) and negative control samples were used as references to distinguish different clusters. In addition, dilutions of synthetic SARS-CoV-2 RNA were also included as controls and analyzed across different PCR cycles and primer pool conditions.

**Viral mutation assessment**. To remove PCR and sequencing errors for the assessment of viral sequence variations, we determined the top enriched amplicon sequence. For this, firstly, paired end reads were stitched together to evaluate full length amplicons. The last 12 nucleotides of read1 sequence are used to join the reverse complement of read2 sequences. No mismatches were allowed for stitching criteria. The number of full length reads per unique sequence variation were counted for each amplicon per sample by matching the 10 nucleotides from the 3′ and 5′ end of the sequence with gene-specific primers. (scripts are available at https://github.com/seda-barutcu/FASTQstitch[27], and https://github.com/seda-barutcu/MultiplexedPCR-DeepSequence-Analysis[28]). The top enriched sequence variant from each sample is used for multiple alignment analysis using CLUSTALW V2.1.

**Non-specific amplicon assessment**. Single-end reads that contain the first 10 nucleotides of the illumina adaptor sequence were counted and binned into relevant forward and reverse gene specific primer pools by matching the first 10 nt of the reads with primer sequences. Relative abundance of the non-specific amplicons was quantified as percentage of the reads corresponding to non-specific amplicon per forward or reverse primer (scripts are available at https://github.com/seda-barutcu/MultiplexedPCR-DeepSequence-Analysis[28]).

**Reporting summary**. Further information on research design is available in the Nature Research Reporting Summary linked to this article.

## Data availability
Data that support the findings of this study have been deposited in the Gene Expression Omnibus (GEO) at NCBI with the accession code GSE160036. Figure 1 raw data, PoC cohort: GEO accession number GSE160031, Fig. 2 and Supplementary Fig. 2 raw data, Test cohort: GEO accession number GSE160032, Fig. 3 and Supplementary Figs. 3, 4 raw data, Pilot cohort: GEO accession number GSE160033, Fig. 4 and Supplementary Figs. 5 and 6 raw data, Extended cohort: GEO accession number GSE160034. Severe acute respiratory syndrome coronavirus 2 isolate Wuhan-Hu-1, complete genome: NCBI sequence ID: NC_045512 was used as reference for primers design and sequence analysis. Source data are provided with this paper.

## Code availability
We provided the code for demultiplexing and mapping at https://github.com/UBrau/SPARpipe[25], quality filtering at https://github.com/UBrau/ModelPerformance[26], viral mutation assessment and non-specific amplicon assessment at https://github.com/seda-barutcu/FASTQstitch[27] and https://github.com/seda-barutcu/MultiplexedPCR-DeepSequence-Analysis[28].

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

## Acknowledgements
The authors thank Drs. Rita Kandel and Jim Woodgett for discussions. The authors are grateful to Tanja Durbic and Kyle Turner in the Donnelly Sequencing Center for sequencing samples and Kathy Fung at the Network Biology Collaboration Center. This work is supported by the Toronto COVID-19 Action Initiative (TCAI) Fund from the University of Toronto awarded to J.L.W., L.P., B.J.B. and R.B., and by a donation from the Krembil Foundation (SHSF Krembil SARS-COV-2) to J.L.W., L.P., and R.B.

## Author contributions
These authors contributed equally: M.-M.A., J.J.H., S.B. J.L.W., M.-M.A., J.J.H., H.H., U.B., B.B. and S.B. designed the study. M.M.A, J.J.H. performed C19-SPAR-Seq experiments. S.B. performed non-specific amplicon/mutation analysis, S.B. and U.B. performed NGS analysis and established C19-SparSeq interpretation pipeline, H.H. performed unsupervised clustering, M.-M.A. and J.J.H. assisted with the rest of the analysis. D.T. collected the samples and purified the RNA. J.P.D. performed qPCR control studies ('BGI') under supervision of R.B. K.C. performed sequencing. M.B. and M.J. assisted with automation optimization. S.L.P., J.K., and S.S. prepared control samples under supervision of L.P. and L.A., T.M., S.P, C.B., and B.H. provided access to patient samples, collection of diagnostics information, and assembly of the cohorts. All experiments were carried out under the supervision of B.J.B, L.P., and J.L.W. The manuscript was written by M.-M.A., J.J.H., S.B., U.B., L.P., and J.L.W. with input from R.B., T.M., S.P., L.A., H.H., and B.J.B.

## Competing interests
J.L.W. is founder and CEO of iTP Biomedica Inc, which employs whole transcriptome NGS tests in cancer, and he is founder and consultant for Fibrocor LP, which is developing therapeutics for fibrotic disease. The other authors declare no competing interests.
