## [Peer Review File · Nature Communications]

Reviewers' Comments:

Reviewer #1:

Remarks to the Author:

This work is an important contribution for testing strategies in the current COVID-19 pandemic. I have only some suggestions regarding the interpretation of results and some questions regarding mapping.

1) Although the results are very promising, there are too bad values in the results: A lower CI-limit of 0.73 for sensitivity (meaning that 27% of infected individuals could potentially not be identified in the worst case) and 0.85 for PPV (meaning that in the worst case, 15% of positive tests are false positives). Both should be discussed.

2) The authors report an "optimum" cut-off of 116 viral reads based on optimum PPV and NPV. More typically, in ROC curve analysis, is to use the Youden-Criterion which is to maximize sensitivity+specificity-1. How does the optimization with respect to PPV and NPV work? Normally I would expect some tradeoff between both, i.e. when gaining in the PPV one will lose in the NPV.

3) What is the biological rationale behind using a cut-off? Of course, I understand, if classification performance becomes better, it's ok. However, would we not assume that if the viral read count is > 0 that there is evidence for an infection?

4) What is the reason why do the negative patients (n=88) have viral reads at all (range: 0-55)?

5) The authors provide confidence limits in Fig 2 and 3, but not in Fig. 4.

6) Could mapping or mutation assessment be improved by using assembled contigs instead of reads? As is for example done in most computational virus detection pipelines.

7) How sensitive would the results be if an individual is infected by another coronavirus? Could the mapping distinguish reads e.g. from SARS-CoV-2 and SARS-CoV-1?

Reviewer #2:

Remarks to the Author:

In the current study, Aynaud and colleagues present a multiplexed, NGS platform for high-throughput detection of SARS-CoV-2. The methodology the authors employ here allows for the analysis of 10's of thousands of samples simultaneously. On a trial cohort of 600 patients, the assay performed with 100% specificity and 91%/>95% sensitivity on low and high viral loads, respectively. The cost for this assay, when used in bulk, is comparable to qPCR on a per sample basis. This technology is timely and its implementation could aid in the effort to identify infected individuals. Overall, the manuscript is largely clear as written and the conclusions are supported by the data.

Comments:

1. Although the authors have included a link for the scripts used in this study, a brief description of the steps involved would be very helpful here.

2. An indication of the computing requirements for this assay would be helpful for those considering adoption of this methodology.

3. For the thresholding, is the total coverage across all amplicons used for this calculation? This is not entirely clear from the text. A more thorough explanation is warranted.

RESPONSE TO REVIEWERS

We would like to thank the two referees for their enthusiasm, judicious comments and thoughtful suggestions about our work. Below is our detailed point-by-point response. Our responses are *in blue* and the original comments are *in black*. To assist the reviewers in assessing the revised version of our manuscript, we have highlighted in yellow areas in the main text that were modified in response to the comments. The referee's reports have been very helpful, and we hope that the reviewers will find our revised manuscript suitable for publication in *Nature Communications*.

Reviewer #1 (Remarks to the Author):

This work is an important contribution for testing strategies in the current COVID-19 pandemic. I have only some suggestions regarding the interpretation of results and some questions regarding mapping.

We thank the reviewer for critical reading of our manuscript and giving insightful input, as well as raising important questions. Please find our responses to each of the questions below.

1) Although the results are very promising, there are to bad values in the results: A lower CI-limit of 0.73 for sensitivity (meaning that 27% of infected individuals could potentially not be identified in the worst case) and 0.85 for PPV (meaning that in the worst case, 15% of positive tests are false positives). Both should be discussed.

In this cohort two of the positive samples highlighted below in Fig. 2b (the cohort that has a lower CI limit of 0.73 highlighted by the reviewer), have qPCR-Ct values from the clinical diagnostic labs that are 38 and 40 in only one tested SARS-CoV-2 gene (N gene). Since values of Ct-38 is the standard upper limit for reliably detecting a positive sample in a qPCR setting, these samples clearly have extremely low levels of RNA. Consistent with this, they were also negative for *RdRP* and *E* (so they are not plotted in the *RdRP* and *E* curves on the left panel graph in Figure 2b). Indeed, it is debatable if these are in fact positive samples, as values in the Ct38-40 range are often considered indeterminant, and some qPCR tests require positivity in at least two genes. Nevertheless, they are called as positive in our test development cohort so we must include them as such, and it is these two samples that scored below the cut-off for calling a positive and account for the lower CI value calculated in Fig. 2d. We have now added new text on page 6 in the manuscript to address this issue.

We would also like to point out that patient samples with such extremely low viral loads are a small percentage of the SARS-CoV-2+ population as assessed in a large-scale evaluation of viral load conducted in another study (Jacot *et al.*, 2020). In fact, our test performance was much better (sensitivity ~0.97) when we projected our sensitivity for samples onto the population distribution of viral loads determined by Jacot *et al.* using ~4000 positive samples (see last section of manuscript). In the revised manuscript we have further elaborated on this important point (page 13).

2) The authors report an “optimum” cut-off of 116 viral reads based on optimum PPV and NPV. More typically, in ROC curve analysis, is to use the Youden-Criterion which is to maximize sensitivity+specificity-1. How does the optimization with respect to PPV and NPV work?

We initially considered the cut-off using Youden-Criteria which is a commonly used method for defining optimum cut-off, as the reviewer pointed out. For the same dataset that yielded the 116 cut-off by PROC, Youden gave an optimum cut point of 26, with a sensitivity of 0.96, specificity of 0.95, and 0.96 accuracy. Although we obtained an increased sensitivity with Youden, compared with PROC results, there was a corresponding drop off in specificity, and accuracy was lower. We now added this information to the manuscript on page 6.

Normally I would expect some tradeoff between both, i.e. when gaining in the PPV one will lose in the NPV.

Yes, this is correct that gaining PPV ($\text{true positive}/(\text{true positive} + \text{false positive})$) would result in a decrease in NPV ($\text{true negative}/(\text{true negative} + \text{false negative})$) and *vice versa*. Depending on the clinical set-up and the priority of sensitivity versus specificity, the threshold applied to C19-SPAR-Seq could be adjusted for better positive or negative case detection as required. We incorporated a discussion of this important point in the revised manuscript on page 6.

3) What is the biological rationale behind using a cut-off? Of course, I understand, if classification performance becomes better, it's ok. However, would we not assume that if the viral read count is > 0 that there is evidence for an infection?

The reviewer is correct, theoretically >0 read counts should indicate positivity. However, because of the inherent technical background noise that most multiplexed high-throughput sequencing platforms generate, we could observe sporadic, trace viral reads in negative-diagnosed samples and technical negative control samples. This could be caused by PCR/sequencing errors, that yield de-multiplexing errors, or trace cross-contamination. We therefore employed a threshold to control for these background events.

4) What is the reason why do the negative patients (n=88) have viral reads at all (range: 0-55)?

As detailed in the above point, the detection of viral reads in negative samples is likely caused by rare background PCR/sequencing errors that result in trace amounts of demultiplexing errors and/or sporadic cross-contamination events.

5) The authors provide confidence limits in Fig 2 and 3, but not in Fig. 4.

We apologize for this oversight. We have now included confidence limits in Fig. 4. For the reviewer's convenience please see the table below detailing the new information added in Figure 4 panel a:

	Value	lower limit	upper limit
Sensitivity	0.857	0.785	0.91
Specificity	1	Na	Na
PPV	1	Na	Na
NPV	0.967	0.951	0.981

6) Could mapping or mutation assessment be improved by using assembled contigs instead of reads? As is for example done in most computational virus detection pipelines.

This is a great suggestion that could be implemented in future versions. In our current testing pipeline, which is optimized for virus detection, coverage of viral sequences is limited to ~ 100 base pair amplicons of 3 independent regions (see Fig. 1), which have been optimized to reduce non-specific amplifications, maximize throughput per run and minimize cost per sample. Future implementation of C19-SPAR-Seq could incorporate panels of multiple overlapping amplicons to allow for contig assembly. In addition, the regions we have designed amplify key residues in the receptor-binding domain and polybasic site required for proper Spike function, as well as capture identified variants, such as the newly emergent N501Y and P681H variant.

7) How sensitive would the results be if an individual is infected by another coronavirus? Could the mapping distinguish reads e.g. from SARS-CoV-2 and SARS-CoV-1?

While the RdRP-specific primers could produce an amplicon from SARS-CoV-1, the Spike-specific primers, targeting the RBD and Polybasic site regions are unique to SARS-CoV-2. In addition, the sequences of these regions do indeed distinguish different CoVs, and our pipeline for detecting mutations would readily identify large-scale sequence variations that would be associated with a new SARS-CoV. We added this information to the manuscript on page 4.

Reviewer #2 (Remarks to the Author):

In the current study, Aynaud and colleagues present a multiplexed, NGS platform for high-throughput detection of SARS-CoV-2. The methodology the authors employ here allows for the analysis of 10's of thousands of samples simultaneously. On a trial cohort of 600 patients, the assay performed with 100% specificity and 91% / >95% sensitivity on low and high viral loads, respectively. The cost for this assay, when used in bulk, is comparable to qPCR on a per sample basis. This technology is timely and its implementation could aid in the effort to identify infected individuals. Overall, the manuscript is largely clear as written and the conclusions are supported by the data.

We would like to thank the reviewer for critical reading of our manuscript and giving insightful input, which we believe have helped us improve the manuscript.

Comments:

1. Although the authors have included a link for the scripts used in this study, a brief description of the steps involved would be very helpful here.

This is an excellent suggestion. We have now added flowcharts as supplementary figures (Supplementary Fig. 7 and 8 in extended data Figures document page 14-17) that provide a step-by-step explanation of C19-SPAR-Seq thresholding and interpretation pipeline. This is included here for the reviewer's convenience.

2. An indication of the computing requirements for this assay would be helpful for those considering adoption of this methodology.

The most computationally intense step of the analysis pipeline is the Illumina bcl2fastq Conversion Software which de-multiplexes the sequencing reads using their sample-specific barcodes. For this step, the computational requirements are 32 GB RAM and minimum 1Gbit network infrastructure, with a Linux operating system. We have now added this information in the methods section of our manuscript on page 19.

3. For the thresholding, is the total coverage across all amplicons used for this calculation? This is not entirely clear from the text. A more thorough explanation is warranted.

We used total RNA counts across all amplicons (as the reviewer indicated above) for the determination of RNA quality. In addition, for the positive/negative thresholding, we used total reads from only viral amplicons. We now expand on this in the revised manuscript on pages 20 to more clearly communicate this information.

Reviewers' Comments:

Reviewer #1:

Remarks to the Author:

The authors have adequately addressed all issues raised by myself and by another reviewer.

Reviewer #2:

Remarks to the Author:

The authors have sufficiently addressed my concerns in their revised manuscript.